# Sequence Order in the Range 1 to 19 by Chimpanzees on a Touchscreen Task: Processing Two-Digit Arabic Numerals

**DOI:** 10.3390/ani13050774

**Published:** 2023-02-21

**Authors:** Akiho Muramatsu, Tetsuro Matsuzawa

**Affiliations:** 1Institute for Advanced Study, Kyoto University, Kyoto 606-8506, Japan; 2Division of Humanity and Social Sciences, California Institute of Technology, Pasadena, CA 91125, USA; 3Department of Pedagogy, Chubu Gakuin University, Gifu 504-8037, Japan; 4College of Life Science, Northwest University, Xi’an 710069, China

**Keywords:** chimpanzee, Arabic numeral, touchscreen, masking task, working memory, transitive inference, decimal number system, word–letter processing, global–local processing, cognitive tradeoff theory

## Abstract

**Simple Summary:**

This study aimed to teach the numerical sequence from 1 to 19 in the decimal system to six chimpanzees. The participants were three mother–child pairs. The original goal for chimpanzees was to touch the numerals on the display from 1 to 19, in this order. The baseline daily training was twofold: touching the adjacent numerals from 1 to X and X to 19 in ascending order. In two separate ways, chimpanzees succeeded at touching adjacent numerals in the range 1 to 19. Systematic tests assessed four factors: range (1 to 9 vs. 1 to 19), adjacency (adjacent vs. nonadjacent numerals), number of stimuli used (three, four, and five), and memory load (nonmemory vs. a memory task called the “masking task”). All four factors were important. A further test directly compared the performance of chimpanzees with that of human participants using the same apparatus and procedure. Both accuracy and response latency showed that processing two-digit numerals was more difficult than one-digit numerals in both species. A chimpanzee named Pal perfectly mastered the order of two-digit numerals just like humans. The difference between the two species was discussed in terms of species-specific global–local information processing.

**Abstract:**

The sequence of Arabic numerals from 1 to 19 was taught to six chimpanzees, three pairs of mother and child. Each chimpanzee participant sat facing a touchscreen on which the numerals appeared in random positions within an imaginary 5-by-8 matrix. They had to touch the numerals in ascending order. Baseline training involved touching the adjacent numerals from 1 to X or from the numeral X to 19. Systematic tests revealed the following results: (1) The range 1 to 9 was easier than 1 to 19. (2) Adjacent numerals were easier than nonadjacent ones. (3) The “masking” (memory task) caused deterioration of performance. All these factors depended on the number of numerals simultaneously presented on the screen. A chimpanzee named Pal mastered the skill of ordering two-digit numerals with 100% accuracy. Human participants were tested in the same experiment with the same procedure. Both species showed relative difficulty in handling two-digit numerals. Global–local information processing is known to be different between humans and other primates. The assessment of chimpanzee performance and comparison with humans were discussed in terms of the possible difference in the global–local dual information processing of two-digit numerals.

## 1. Introduction

There are many studies on numbers from a developmental and evolutionary perspective. This study aimed to teach the numerical sequence from 1 to 19 in the decimal system to six chimpanzees. They had already learned the numerical order in the range 1 to 9 [1,2] but the present study expanded this from 1 to 19. The original goal for chimpanzees was to touch the Arabic numerals on the display 1 to 19 in this order and understand the decimal number system. To test for a possible species difference in processing the numerals from 1 to 19, chimpanzee performance was directly compared with that of humans using the same task, apparatus, procedure, and place.

Before introducing the research background of the existing literature on numbers, we explain and define some technical terms. “Number” means the concept of numbers. “Numerosity” refers to a property of a stimulus that is defined by the number of discriminable elements it contains [3]. “Numerals” are Arabic numerals, which are the media for representing numbers. For a further detailed explanation of the decimal number system, we also use another term: “digit”. A digit is a single numeral from 0 to 9. Therefore, in the decimal system, a numeral may be represented by two or more digits, such as 12 and 345. The number 12, for example, is referred to as a two-digit (or double-digit) numeral in this article.

In sum, digits and numerals are used to refer to numerosity and to represent numbers. However, it must be noted that digits and numerals may not always have a number concept. For example, suppose that there are three classrooms at a school, such as classes 1, 2, and 3, or A, B, and C. The numerals in this example are not based on any cardinal or ordinal scale of numbers; instead, they are used on a nominal scale of numbers to distinguish three different things. The present study does not directly focus on numbers as it aimed to teach a sequence of Arabic numerals from 1 to 19. Therefore, this study is related to the existing literature on sequence learning and the sequential order of items [4,5,6,7,8,9].

Numbers have been studied in humans and nonhuman animals [10,11,12,13,14]. The human number concept has been studied from developmental perspectives [15,16,17,18,19]. Without conscious counting, human infants can discriminate, represent, and remember a small number of items [20,21,22,23]. Verbal counting may have precursors during infancy based on subitizing, the direct perception of numbers. Beyond subitizing, young children start to count one by one and comprehend numbers as part of their linguistic capability.

The concept of numbers has been studied from an evolutionary perspective as well. Studies on numbers have been carried out with a wide range of species, including invertebrates [24], fish [25,26], pigeons [27,28,29,30], a grey parrot [31,32], rats [33,34,35,36], and monkeys and apes [37,38,39,40,41,42,43]. In addition to behavioral studies, there are neurophysiological studies on monkeys that support the existence of a specific substrate of the number concept in the brain [11,44,45,46].

Chimpanzees and monkeys can provide a unique opportunity for applying various tests in the same situations as humans. Ferster (1964) first used the binary number system in which two chimpanzees learned from 1 (001) to 7 (111) by turning three lights on and off [47]. By introducing Arabic numerals, it is possible to match the chimpanzee studies precisely to studies in humans. Matsuzawa (1985) [48,49] and Boysen and Berntson (1989) [50,51] started to use Arabic numerals to represent numbers. Chimpanzee Ai of the Primate Research Institute of Kyoto University (KUPRI) learned to use Arabic numerals arranged on a keyboard connected to a computer [48]. She learned to use the numerals from 1 to 6 on the keyboard and combined the skill with naming colors and objects shown in a display window. This task is referred to as symbolic matching-to-sample (symbolic MTS). She succeeded to press the corresponding keys for “red” + “pencil” + “5” in sequence for describing five red pencils that were shown to her. Ai mastered the skills of both ordinals and cardinals, and coding and decoding [52,53,54]. The ordinal sequence of numerals was expanded from 1 to 9 [55,56]. Additionally, Ai learned the meaning of the numeral 0 [53]. The other chimpanzees of KUPRI have also learned numbers in various tasks in this line of studies [57,58,59,60].

The major findings of earlier studies on numbers in nonhuman primates (mainly chimpanzees and macaques) can be summarized with the following three points. First, they can master the skill of using Arabic numerals. Second, they may use the numerals for both cardinality and ordinality and in both productive uses and receptive uses. Third, there have been no studies teaching the use of decimals to systematically increase the repertoire of using numerals beyond 9.

What kind of numerosity judgment is used by animals: subitizing, counting, or magnitude estimation? This is not clear, because the numerical repertoire of the animals is still small, with a few exceptions [42,43]. One solution is to extend the numerical sequence as a larger repertoire, as a first step for further study following numerosity judgment. With respect to previous efforts, the present study is unique in being the first attempt to expand the numerical sequence from one-digit numerals to two-digit numerals up to 19. It should be stressed that this study is about the sequential order of the numerals, not numerosity judgment. The task is not about cardinality, such as when “counting” objects. Sequential learning of numerals may provide the basis for the psychophysical sense of numerosity and the corresponding use of symbolic numbers [61]. Young human children recite numerical sequences, saying “one-two-three-four-five-six-seven-eight-nine-ten”, and so on, without fully understanding the meaning. The sequence of numerals may play a fundamental role in the future understanding of the number concept. The present study aimed to show this kind of *precise numerical sequencing* in chimpanzees.

Numbers can be described in various ways. For example, the number “ten” can be described in the binary system as 1010. In the octal system, it is 12. In the decimal system, it is 10. In the duodecimal and hexadecimal systems, it is A. The decimal number is a structuralized notation system: there is a “spiral staircase” or a “clockwise” structure. Each stair of the single-digit numerals goes up and round to the next stair of two-digit numerals from 10 to 19, to further stairs from 20 to 29, from 30 to 39, and so on. The question is whether the chimpanzee can master this kind of ordering system of decimal numerals.

The present study aimed to teach three mother–child pairs of chimpanzees to use the numerical sequence 1 to 19 and to understand the notation system of the decimal. The participants had already learned the numerical sequence of 1 to 9 [1,2] and one step forward to the sequence from 10 to 19 [58]. Based on the previous research, the present study aimed to establish a way to examine and evaluate chimpanzees’ learning of 1 to 19 in the decimal system. The performance of processing two-digit numerals was directly compared between humans and chimpanzees in the same tests with the same apparatus and procedure. This cross-species comparison was planned to clarify the underlying mechanism for processing two-digit numerals. The present study postulates that the key issue is global–local dual information processing in humans and nonhuman animals [62,63,64,65,66,67].

The well-known phenomenon called the Navon effect [62] was discovered by David Navon, who measured the speed at which people process global and local information. When objects are arranged in groups, they possess global features and local features. For example, a group of trees has local features (the individual trees) and the feature of a forest (the trees together). In this framework, a group of digits (two-digit numerals) has local features (the individual digits) and a global feature (the digits together make a numeral that has the real meaning). Humans are faster at identifying features at the global than at the local level (in other words, they show global precedence). In contrast, the existing literature on nonhuman primates (chimpanzees [68,69], baboons [69], and capuchin monkeys [70]) suggests they identify the local level faster than the global. Thus, this study is a pilot study to explore possible evolutionary origins of the difficulty in processing two-digit numerals.

## 2. Materials and Methods

### 2.1. Participants: Chimpanzees

The participants were three mother–child chimpanzee pairs. The study period lasted three years and eight months, from April 2011 to November 2014. All three children were 10 years old at the beginning and reached 14 years of age by the end (Table 1). They were not fully independent from their mothers, so the mother–child pairs came together to the test booths. During the present study period, the pairs came to the booth as follows: 630 days in total for Ai–Ayumu, 477 days for Chloe–Cleo, and 498 days for Pan–Pal.

In prior work, all six participants had learned the sequence from 1 to 9 [1,2]. An adult female, Ai, had learned the meaning of 0 as well, so her range was 0 to 9. She also mastered skills in the ordinal and cardinal aspects of numbers [53,54]. The other five chimpanzees had experience touching the numerals in ascending order, but no prior experience using the numerals for the cardinal task. As the first step toward 1 to 19, all six chimpanzees gained experience in generalizing the skill to the set of numerals from 10 to 19 by introducing the two adjacent numerals successively: 9-10, 10-11, 11-12, 12-13, and so on up to 18-19. The initial training procedure on two-digit numerals is described in detail in a separate article [58]. Thus, they were ready to be trained and tested on the numerical sequence from 1 to 19 in the present study. The rearing condition and the experience of the six chimpanzees are described in Matsuzawa et al. [71].

The present study is a part of a larger project studying chimpanzee cognition at KUPRI [71,72]. The participants experienced different kinds of cognitive experiments in parallel during this study period [73,74,75,76,77,78,79,80,81,82,83]. Please refer to the details in Section A.1.

In general, KUPRI chimpanzees had a maximum of seven feeding opportunities in a day: four laboratory tests plus three daily meals. The KUPRI schedule included two series of sessions in the morning and another two in the afternoon for cognitive tests. One series of sessions either in the morning or in the afternoon was allocated to each pair for the present study. This arrangement simulated the daily cycle of natural feeding behavior in wild chimpanzees [84]. The chimpanzees were free and spent the time between sessions in the enriched outdoor enclosure with trees, shrubs, and a stream [85]. When the present study started, the chimpanzee group had 14 members [71], including 3 generations of chimpanzees of patrilineal lineage, similar to a wild chimpanzee community. The three children had grown up in this socially enriched environment. All chimpanzees understand their names [86]. The experimenters called the name of a particular chimpanzee to invite them into the test booth. The chimpanzees were completely free to choose whether or not to participate in tests. However, as this was a daily routine, all chimpanzees were willing to come to the booth. This study followed the Guideline of Care and Use of Nonhuman Primates, KUPRI, and was approved by the Animal Welfare and Care Committee of KUPRI (see Ethics statement).

### 2.2. Participants: Humans

There were 6 human participants, 24–29 years old, including both sexes (4 females and 2 males). All were right-handers. They were students and staff of KUPRI. We collected the chimpanzee data first and then tested the humans in 2015 (April to June). The humans were tested with the nonmemory task only (see details of the test procedure in Section 3.4). The task was to touch the numerals on the screen in ascending order. The range was either 1 to 9 or 1 to 19. The number of numerals was 3, 4, or 5. The numerals were adjacent or nonadjacent. There was no memory load, so the task was an easy one for adult humans.

The human participants were naïve to the task but they had plenty of opportunities in the institute and their daily life to use the touchscreen and Arabic numerals. We did not explicitly train the participants to become experts in quickly touching the numerals. Computer use, such as regularly playing computer games, may lead to decreased response latency in some visual/motor tasks; however, this was not the purpose of the present study, which aimed to evaluate the performance of naïve humans in this setting of two-digit numerals without specific training (see Ethics statement).

### 2.3. Stimuli

The stimuli were 19 Arabic numerals from 1 through 19. The font was MSP Gothic. They appeared on a CRT monitor touchscreen. The numerals were displayed as white stimuli on a black background. The height of the numerals was 3.5 cm. The resolution of the monitor was 1023 by 768 pixels. The numerals appeared randomly in one of the 40 imaginary positions in a matrix of five rows and eight columns. There was a so-called start key to initiate a trial: it was a white circle that appeared at the sixth bottom row of the CRT.

### 2.4. Number, Numerosity, Numeral, and Digit

As we described in the Introduction, there are various related and confusing terms to describe the study of numbers. Please see the definition in the Introduction about “Number”, ”Numerosity”, “Numeral”, and ”Digit”.

### 2.5. Apparatus

A “touchscreen” has an input device called a “touch detector or touch panel” and an output device called a “monitor or screen” [87,88]. A touch by the participant was detected by a touchscreen (Mitsubishi Electric Engineering 15-inch LCD touchscreen monitor: TSD-FT157-MN and TSD-AT1515-MN, Tokyo, Japan). As shown in Figure 1, the touchscreen was encased in a translucent acrylic box and set just behind a translucent panel which prevented the chimpanzee from strongly banging the screen. The participant touched the screen through a window opened at the lower part of the box. For an adult chimpanzee sitting in front of the monitor, the center of the monitor was at eye level. The distance from the eyes to the monitor was about 30 cm. The history of this apparatus is described in Appendix A S5. To test a mother and child at the same time, we used a “twin booth” consisting of two identical booths located side by side. Each booth was 1.8 m by 1.8 m by 2.0 m in height. Spontaneously, all three mothers came to the far side of the twin booth and the three children chose the near side to the entrance. Human participants were tested by using the same apparatus following the same procedure at the same place, but alone.

The task was controlled by standard PC-type computers running under Windows XP© and Windows 7 © OS. The program controlling the experimental session was made with Visual Basic © 6.0. The entire experiment was preprogrammed with the previously determined stimulus sequence and position sequence files. The food reward was delivered by an automatic feeder (Bio Medica: BUF-310-P50, Osaka, Japan) connected to the computer. The feeder had a disc of 50 small compartments, and a brush rotation automatically delivered a piece of food to the place just below the touch panel. Thus, daily cognitive test sessions were fully automated in preplanned ways and with no interference by experimenters. This means that there was no social cueing.

### 2.6. General Procedure

To come to the experimental booth, the chimpanzees walked through a corridor about 50 m long which connected the outdoor enclosure to the booth. Each trial went as follows. A white circle appeared on the monitor. When the participant touched it with their fingers, the circle disappeared, and several Arabic numerals immediately appeared in random positions on the monitor. There were 5 by 8 imaginary matrix positions on the monitor in which each numeral appeared. The correct response was to touch the ‘smaller’ number first, followed by the ‘larger’ number(s). CRF (Continuous Reinforcement Schedule) was applied: every correct response was rewarded. The correct choice was signaled by a chime and followed by automated food delivery. The food reward was a piece or a half-piece of raisin or a very small piece of apple (1.5 g per piece on average). A wrong response was signaled by a buzzer sound and followed by a 3 s blackout and a return to the start key. The inter-trial interval (ITI) was set at 1 s. After the ITI, the next trial started: a white circle appeared on the monitor. A session consisted of 50 trials without exception in all training and assessment tests. It was completed in 3 to 5 min. The inter-session interval was 1 to 3 min. Participants received four to six continuous sessions each day on average. After completing the 30 to 60 min series of sessions for the cognitive tests, the participants rejoined the other chimpanzees in the outdoor enclosure (see Figure 2).

Before starting each session, the experimenter ran a Q and A with the computer program to set up the task parameters. Parameters of a session were as follows: the number of trials in a session (fixed at 50 trials), ITI (fixed at 1 s), fixed ratio (FR) for reinforcement (fixed at FR1), time out (fixed at 3 s blackout), correction trial (no correction trial; an error trial did not repeat and the next new trial started), sequence files of stimulus presentation and the position of numerals appearing in the imaginary 5-by-8 matrix on the screen (All conditions of stimulus and position were randomized in a session).

## 3. Method: Baseline Training and Assessment Tests

### 3.1. Baseline Training of Touching Adjacent Numerals: VarNumMix (VNM) Task

#### 3.1.1. VNM-Startfix Task

During the present study, chimpanzees received daily training in numerical ordering using the touchscreen system. The numerals appeared on the screen and the task was to touch them in ascending order. The task was named the “VarNumMix (VNM)”, in which adjacent numerals appeared in every trial, although the number of numerals varied in each trial.

Each trial was unique and randomized within a session. For example, the “VNM 1 to 14” task means that a trial could be either 1, 1-2, 1-2-3, 1-2-3-4…or 1-2-3-4-5-6-7-8-9-10-11-12-13-14. It must be noted that all of the numerals were randomly scattered on the screen (see Figure 1). VNM tasks of the present study started from “VNM 1 to 9” to “VNM 1 to 10”, “VNM 1 to 11”, “VNM 1 to 12”, and so on, step by step. This sequence in the VNM task was characterized by the “Startfix” condition in which the numerals always started from the numeral 1. Therefore, this is specifically called the “VNM-Startfix” task hereafter.

#### 3.1.2. VNM-Endfix Task

In another type of VNM task, the end of the sequence was always fixed as the numeral 19. Thus, this is called the “VNM-Endfix” condition. For example, “VNM-Endfix 13 to 19” means that a trial could be either 19, 18-19, 17-18-19, 16-17-18-19…and so on up to 13-14-15-16-17-18-19. This training focused on teaching the end part of the long numerical sequence from 1 to 19. In the present study, we started with “VNM-Endfix 16-19”. After reaching the criterion of 90% accuracy in a 50-trial session, the task became one step more difficult, meaning VNM-Endfix 15 to 19, and so on.

#### 3.1.3. Baseline Training to Maintain Motivation

The baseline training of VNM tasks was characterized by adjacent numerals. In both Startfix and Endfix conditions, chimpanzees learned to touch the adjacent numerals in ascending order. Nonadjacent numerals were not used in the training but were used for further assessment tests.

This training method of VNM tasks aimed to keep accuracy and motivation high by reducing the difficulty of the task. It is very important to keep motivation high for chimpanzees to participate in cognitive tasks. For example, suppose that all 19 numerals appeared scattered across the screen at one time and the chimpanzees were asked to touch them in ascending order. The chance level of the correct order is _19_P_19_ equal to one out of 1.21645 × 10^17^, which is approximately *one out of some quadrillions*, too much for some chimpanzees in the training phase, which could easily lead to loss of motivation to participate. Thus, the present study used the VNM training method to mix difficult trials with easy ones within a session.

#### 3.1.4. Baseline Training to Assess Daily Fluctuations 

Baseline training occurred each day. The chimpanzees received baseline training on the VNM task whenever they came to the booths. This was done to improve their performance at ordering numerals while evaluating daily fluctuations in performance in each individual. In general, the criteria for moving to the next condition were kept constant, i.e., more than 90% correct in a session. However, in some chimpanzees, the criterion gradually shifted to 85% and then 80% accuracy in a session, depending on their performance. The whole experiment was designed to make the tasks gradually more demanding based on each individual’s daily performance (see Section 4.1 and Section 4.2).

### 3.2. Assessment Tests: Range, Adjacency, Number of Numerals, and Memory

In parallel to the baseline daily training, chimpanzees underwent systematic tests to assess their understanding of numerical sequence (Figure 3). The assessment tests aimed to evaluate the progress of learning the numerical sequence of 1 to 19. The tests were carried out under the condition of differential feedback (positive reinforcement training); a correct answer was rewarded with food and an error was not. Therefore, in terms of the reinforcement history, it was an assessment test and also training on the numerical sequence by differential feedback.

In the assessment tests, we focused on four factors that could be influencing performance on numerical ordering. First, the range of numerals was either 1 to 9 or 1 to 19. Second, the adjacency was either adjacent numerals or nonadjacent ones. Third, the number of numerals tested was 3, 4, or 5. Fourth, the ‘memory’ refers to whether the task was a nonmemory (nonmasking) task that required no memory at all or a task that required memorizing of numerals. In the memory condition, after touching the ‘smallest’ numeral, the other numerals were immediately masked by a black-and-white checker pattern. Therefore, the chimpanzees had to remember which numeral appeared in which position of the monitor before starting the touch. The memory condition is called the “masking task” [1,89]. All chimpanzees were tested about factors of adjacency and memory in the range 1 to 9 [1,2]. They underwent some sessions for masking tasks in the range 1 to 9 during the present study to assess any aging effect on working memory. However, the two-digit numerals in the range 1 to 19 were only used in the present study.

The four factors were systematically introduced in the assessment tests. For example, in the test condition “5, nonadjacent numerals, in the range 1 to 19, with nonmemory”, five numerals, such as 5-9-12-13-19, were scattered in random positions on the screen. There were four factors: range, adjacency, the number of numerals, and memory, giving 2 × 2 × 3 × 2 = 24 conditions. Each condition was tested in one session of 50 trials. The order of testing the 24 conditions was not fully randomized but moved from easier to more difficult ones. The assessment test was conducted twice to confirm performance stability (September 2013 and March 2014, a 6-month interval).

### 3.3. Assessment of Adjacent Numerals Including the Numeral 10

In addition to the four factors, there is a unique problem with numerals in the decimal number system: the carryover at the numeral 10. We gave an assessment test to evaluate the difficulty of processing the numeral 10 in chimpanzees. The task was to touch four adjacent numerals in the range 1 to 19. Thus, there were 16 patterns in this task: 1-2-3-4, 2-3-4-5, 3-4-5-6, and so on up to 16-17-18-19. One of them was randomly presented in a session of 50 trials. We ran this test 10 times, for a total of 500 trials for each chimpanzee. Thus, each of the 16 patterns was tested 31 or 32 times. We collected the data for each pattern at the end of the study period. This is an assessment test and also a part of the intensive training of adjacent numerals including the numeral 10.

### 3.4. Comparison of Humans and Chimpanzees in Terms of Accuracy and Response Latency

Chimpanzee performance was directly compared with human performance. All six chimpanzees received the assessment tests for four factors. However, two adult chimpanzees, Chloe and Pan, did not master the sequence from 1 to 19 completely. This means that the VNM-Startfix task and VNM-Endfix task did not cover the entire range of the numerals from 1 to 19. Therefore, these two chimpanzees were excluded from further tests for human–chimpanzee comparisons, leaving the four chimpanzees Ai, Ayumu, Cleo, and Pal, who mastered the skill of ordering the numerals in the range 1 to 19. They were compared with six humans (H1, H2, H3, H4, H5, and H6) who received the same test. Chimpanzee data were the same as a part of the assessment test. Human data were collected under the same procedures as chimpanzees.

The chimpanzees and humans were compared in the condition of nonadjacent numerals. There were 12 conditions in total. The present study focuses on one condition among them, namely “four nonadjacent numerals in the range 1 to 19 with a nonmemory task”. In this task, the four numerals were randomly chosen from 1 to 19. Therefore, there are many combinations of four numerals: _19_C_4_ = 3876. The 3876 patterns can be classified into five groups. The first group is labeled as “Under10”, which means that four numerals are all one-digit numerals such as 2-5-6-9. The second group is “Cod1”, which means that a two-digit numeral is included as one of the four numerals such as 1-5-7-13. The third group is “Cod2”, which means that two two-digit numerals are included as two of the four numerals such as 5-7-12-18. The fourth group is “Cod3”, which means that three two-digit numerals are included such as 3-10-15-19. Finally, the fifth group is “Over10”, which means that four numerals are all two-digit numerals such as 11-14-17-18. The major difference among the five groups is the total number of digits to be processed. The more two-digit numerals, the more digits to be processed. Note that the total number of digits that appeared on the screen increased from four, five, six, and seven to eight along with the five conditions: from all one-digit to all two-digit conditions.

The human participants received the verbal instruction: “Please touch the numerals in ascending order”. There was no instruction to make quick decisions. Humans and chimpanzees were tested at the same place using the same apparatus and following the same procedure. The differences were no verbal instruction for chimpanzees and no food reward for humans.

### 3.5. Summary of Methods

To sum up the Methods, Table 2 describes all tasks described in Section 3. The list includes a basic description of the task and which chimpanzees received the task. It shows short names for the tasks, which are referenced throughout the text. The tasks are ordered in increasing complexity. The table is provided for looking up the different tasks while reading the manuscript. Concerning the comparison of the two species, three points should be noted. First, chimpanzees were trained to touch two-digit numerals in VNM tasks, whereas humans received no such training because they were already familiar with this response. Second, chimpanzees were assessed on four factors, compared to three factors for humans (memory load not tested). The two species were compared in the same condition of nonadjacent numerals. Third, only chimpanzees were assessed for the effect of memory load, intensively surveyed for the role of the numeral “10”, and repeatedly assessed on tests on four factors. These tests are designed to identify the difficulty of processing two-digit numerals for chimpanzees. Details of the training method and metadata are summarized in Section A.2.

## 4. Results

### 4.1. Baseline Training of Touching Adjacent Numerals: VarNumMix (VNM) Task

#### 4.1.1. Summary of the VNM Tasks: Accuracy of Each Chimpanzee in Each Stage

All six chimpanzees were trained to touch numerals from 1 to 19 in ascending order. The baseline training was to touch the adjacent numerals on the screen from 1 to X. X is the maximum number in the sequential ascending order. As described in the Methods section (Section 3.1), the task was called the “VarNumMix task (VNM in short)”, in which each trial was varied in terms of the number of numerals. The task of “VNM-Startfix 1 to 16” means that each trial always started from the numeral 1 (which means the start numeral was fixed) and could be either 1-2, 1-2-3, 1-2-3-4, and so on, or 1-2-3-4-5-6-7-8-9-10-11-12-13-14-15-16 within a session of 50 trials. There was also the “VNM-Endfix” task. Table 3 summarizes each chimpanzee’s accuracy in each stage of training.

#### 4.1.2. VNM-Startfix Task

The baseline training of the VNM-Startfix task continued up to 1 to 16 for chimpanzee Ai. Figure 4 shows Ai’s performance as the representative. Accuracy gradually dropped as a function of the number of numerals to be processed. Ai reached the level of sequentially touching numerals from 1 to 16, but not more. We stopped at this point to avoid too much pressure on the chimpanzee. The other chimpanzees followed the same pattern. Ayumu reached the stage of 1 to 18, and Pal reached 1 to 17. Data for each chimpanzee and the average performance of the six chimpanzees are shown in Table 3 and Section A.3. See the trials in the Appendix A.

#### 4.1.3. VNM-Endfix Task

We introduced the “Endfiix condition” in addition to the training with the “Startfix condition”. This task tested adjacent numerals from X to 19 to teach the end part of a long sequence. The task always ended with the numeral 19. Figure 5 shows the performance of Ai as a representative. She started with the task of “VNM-Endfx 16 to 19” and ended with “VNM-Endfix 9 to 19”. The longest numeral of the “VNM-Endfix 9 to 19” is 12 numerals (9-10-11-12-13-14-15-16-17-18-19). We stopped the baseline training of VNM-Endfix for Ai at this point. The other chimpanzees followed similarly. The data for each chimpanzee and the average performance of the six chimpanzees are shown in Table 3 and Section A.3. See the trials in the Appendix A.

#### 4.1.4. Comparison of Startfix Task and Endfix Task

In the Baseline training, the two tasks, VNM-Startfix and VNM-Endfix, helped the chimpanzees to learn sequential touching from 1 to X and also from X to 19. The two tasks were complementary to each other but not equal. The Startfix condition was easier than the Endfix condition for all chimpanzees because the Startfix condition had been trained for a long time since the beginning of training the chimpanzees on numeral orders. Another reason is in the starting numeral of each trial. In the Startfix condition, every trial started from the numeral 1, so the chimpanzees had to find the numeral 1 among the numerals scattered on the screen. There was no varied condition, but the numeral 1 was always found on the screen. In contrast, in the Endfix condition, every trial changed the ‘smaller’ number while keeping the last numeral 19 fixed. It was difficult for the chimpanzees to find which was the ‘smaller’ number on the display in this case because it always changes from one trial to the next.

Figure 6 shows the comparison of Ai’s performance in the two conditions. There is an overlapping zone for 9, 10, and 11 numerals. Performance on the Startfix condition (94% on average) was better than the Endfix condition (76% on average) in the corresponding overlapping zone. Although the Endfix condition was demanding, the chimpanzees mastered the task. This means that the chimpanzee can find the ‘smaller’ number even though there are many numerals simultaneously on display. Each individual’s data are shown in Section A.3 (Table A1), and the average data are shown in Section A.3 (Figure A1).

#### 4.1.5. Best Performance in Touching Numerals in the Range 1 to 19

The other chimpanzees’ data showed a similar pattern to Ai. The longest sequence tested were as follows: Ayumu, 1 to 18 and 8 to 19; Pal, 1 to 17 and 9 to 19; Ai, 1 to 16 and 9 to 19; Cleo, 1 to 15 and 10 to 19; Chloe, 1 to 13 and 13 to 19; Pan, 1 to 12 and 14 to 19, respectively.

Five of the six chimpanzees show overlap in the range 1 to 19: Ai, Ayumu, Chloe, Cleo, and Pal: only Pan failed to reach this level. They could touch the two-digit numerals in the range 1 to 19 if the sequence was divided into two parts. 

Ayumu touched 1 to 17 precisely in this order (see Figure 1), which was the best performance obtained so far in processing two-digit numerals in one trial.

### 4.2. Range, Adjacency, Number of Numerals, and Memory

#### 4.2.1. Four Factors

As described in the Methods section (Section 3.2), four factors may influence numerical ordering performance: range, adjacency, number of numerals, and memory. Table 4 shows the results of the second assessment test (The results of the first test are in Section A.4 Table A2). The interval between the two tests was about six months. During this time, average performance was slightly improved by about 2% accuracy. The correlation between the two tests was very high: *r* = 0.941. To avoid presenting the same results twice, we describe the data from the second and the final test as follows.

Table 4 shows all individual data in which all four factors had a significant effect. First, for all six participants, the range 1 to 19 was more difficult than 1 to 9. Second, the nonadjacent numerals were more difficult than adjacent ones. Third, the memory task was more difficult than the nonmemory task. Fourth, in all conditions, when the number of numerals increased to three, four, and five, the task became more difficult. There was almost no exception in the conditions of 2 × 2 × 2 × 3 = 24 cases (range × adjacency × memory × number of numerals) in all six individuals. There were some exceptional cases among 144 cells. For example, in three cases Ayumu performed better on the masking (memory) task than the nonmasking (nonmemory) task. For Ayumu, the memory task was as easy as the nonmemory task, due to a sort of ceiling effect (100, 98, and 96% accuracy in a session of 50 trials), alongside inevitable small daily fluctuations in performance.

Although the tendency was the same among these six chimpanzees, there was a clear individual difference in the average performance level. Overall performance in the assessment test was best in Ayumu, with 85% accuracy on average over all 24 conditions. The second best was Pal (79%), followed by Ai (75%) and Cleo (75%), with the remaining two adults, Chloe (61%) and Pan (57%), further behind. There was a correlation between the performance of VNM tasks (Table 3) and the assessment test (Table 4): better learners in training showed better performance on the tests. This result showed that the children outperformed their mothers.

#### 4.2.2. Range

Figure 7 shows the effect of range: comparison of a nonmasking task in the range 1 to 9 and the range 1 to 19 (see data in Table 4). For example, in the five-numeral condition, the display showed five numerals, such as 1-3-5-8-9 or 4-7-12-13-16. The wide range 1 to 19 was more difficult than the narrow range 1 to 9. Performance decreased as a function of the number of numerals: three, four, or five.

#### 4.2.3. Adjacency

Figure 8 shows the comparison of adjacent and nonadjacent numerals in the range 1 to 19 (see data in Table 4). For example, in the five-numeral condition, the display showed five numerals, such as 7-8-9-10-11, in the adjacent condition and 8-10-13-16-19 in the nonadjacent condition. The nonadjacent condition was more difficult than the adjacent one. This tendency was also confirmed in the range 1 to 9 (see Table 4). As we predicted, performance decreased as a function of the number of numerals: three, four, or five.

#### 4.2.4. Memory

Figure 9 shows the effect of memory: comparison of nonmasking and masking tasks in the conditions of range 1 to 19 and nonadjacent numerals (see the bottom six rows of Table 4). For example, in the five-numeral condition, the display showed the five numerals 5-12-13-16-19. In the masking (memory) task, the participant had to remember the five numerals before touching the first one.

Performance decreased as a function of the number of numerals. It also decreased in the nonmasking control task. For the chimpanzees, it was difficult to touch, for example, the numerals 5-8-12-14-19. Therefore, the decrement in performance was not solely due to memory but was partly due to the difficulty of touching five numerals which included two digits in the decimal number system. In short, the memory ability of chimpanzees can be tested with the range of 1 to 19, but a proper control condition is required to subtract the contribution of the nonmemory factor; see Figure 3 and the Appendix A.

### 4.3. Adjacent Numerals Showed Difficulty in Processing the Numeral 10

We gave an assessment test to evaluate the difficulty of processing the numeral 10 in chimpanzees. The task was to touch four adjacent numerals in ascending order. There were 16 patterns in this task: 1-2-3-4, 2-3-4-5, 3-4-5-6, …, 16-17-18-19, randomly presented in a session (see Section 3.3). Figure 10 shows accuracy (% correct) with four adjacent numerals. It shows the accuracy of 16 patterns of four adjacent numerals in the range 1 to 19. We tested four chimpanzees, Ai, Ayumu, Cleo, and Pal, who mastered the numeral order in the range of 1 to 19 by showing good overlapping in the range of numerals in the VNM-Startfix and VNM-Endfix tasks.

There were 16 patterns in this task, which can be classified into three groups. The first is “Group 1 to 9” (Group 1), in which the four adjacent numerals are within the range 1 to 9. There were six patterns from 1-2-3-4 to 6-7-8-9. The second is “Group 11 to 19” (Group 2), in which the four adjacent numerals are within the range 11 to 19. There were six patterns from 11-12-13-14 to 16-17-18-19. The third is in between: “Group including 10” (Group 3), in which the four adjacent numerals include the numeral 10. There were four patterns in this group: 7-8-9-10, 8-9-10-11, 9-10-11-12, and 10-11-12-13.

Average accuracy was 91.7% (SD = 5.5%) in “Group 1 to 9” and 92.1% (SD = 5.6 %) in “Group 11 to 19”. The chimpanzees processed the two-digit numerals (11, 12, 13….19) as easily as the one-digit numerals (1, 2, 3….9) if the numeral 10 was not one of the four numerals. They might, for example, put the left side of the two-digits aside and make the judgment by focusing on the right side. It must be noted that this assessment test was performed for adjacent numerals and not nonadjacent numerals.

In contrast, the average accuracy dropped to 80.9% (SD = 13.3%) in Group 3. If the statistical analysis is applied to the data, it shows a significant difference between the groups: *p* = 0.022 in Group 1 and Group 3, and *p* = 0.020 in Group 2 and Group 3, in the t-test. The chimpanzees mastered the skill of processing the numerals 1 to 19. Even more, the processing of 1–9 and 11–19 was equal in terms of accuracy (not the response latency described later). However, if the numeral 10 was included in the patterns, it was a little difficult for them.

Although the tendency was common to all four chimpanzees, there was a marked individual difference: compared to the other three chimpanzees, Cleo had extreme difficulty in processing the numeral 10. Because of this individual difference, it may not be adequate to apply statistical tests for the group level. However, in addition to Cleo, Figure 10 shows that the sequence of 8-9-10-11 was the most difficult one for the other three chimpanzees too. The accuracy of this sequence was about 80 % correct for the three chimpanzees. It must be also noted that the chance level of touching four numerals in the correct order is about 4%. Therefore, in conclusion, chimpanzee performance was very high even in difficult patterns.

### 4.4. Comparison of Humans and Chimpanzees: Accuracy

The difficulty of processing two-digit numerals (the numeral 10 and more) was shown in humans too. Figure 11 shows the accuracy (% correct) of performance in chimpanzees (*n* = 4) and humans (*n* = 6), showing both the individual data and the average. All participants were tested on nonadjacent numerals in 1 to 19 in the four-numeral condition. For example, the display showed four numerals such as 2-5-7-8, 4-8-9-15, 6-8-12-15, 5-10-12-19, 13-16-17-19, and so on. There are many combinations of four numerals: _19_C_4_ = 3876. Further details of human performance are given in Section A.5 and Appendix A.

Among the 3876 patterns, there are one-digit-only numerals, such as 2-5-7-8, which are labeled “Under10”. Combinations such as 4-8-9-15 are called “Cod1,” which means they contain one two-digit numeral. Combinations such as 6-8-12-15 are labeled “Cod2” (contains two two-digit numerals). Combinations such as 5-10-12-19 are labeled “Cod3” (three two-digit numerals). Combinations such as 13-16-17-19 are called “Over10”: all stimuli are two-digit numerals (see Section 3.4). In the four-numeral conditions, there were 24 possible sequences of four numerals (_4_P_3_), with only one correct order. Therefore, the chance level of correctly touching four numerals was only about 4%. As shown in Figure 11, the performance of chimpanzees was very high; much higher than chance.

Humans are good at touching four numerals in ascending order, with accuracy in the range of 94 to 100%. Humans showed no difficulty in touching four numerals in ascending order if they consisted of either single digits only (Under10) or double digits only (Over10). However, humans made some interesting mistakes in the intermediate conditions (Cod1, Cod2, Cod3) containing mixtures of one-digit and two-digit numerals, except for one individual (H3) who took a very long time to make decisions (see response latency results in the next section). In other words, even for human participants, there was some ‘difficulty’ in touching four numerals in ascending order in the range 1 to 19. This was most marked in Condition “Cod2” with numerals such as 6-8-12-15, in which the participant may have touched “12” first rather than “6”. This kind of confusion can occur even in human adults when judging ascending order of a mixture of one-digit and two-digit numerals in the decimal system. However, humans correctly identified the numerical order if all four numerals are 10 or more (Condition “Over10”). The Over10 condition was as easy as the Under10 condition at least in terms of accuracy. This means a complete understanding of the carry-over of the numeral 10 in this visual discrimination task.

For chimpanzees, it was clear that none had difficulty touching four numerals in ascending order if they consisted of single digits (Under10). All four participants showed 100% accuracy. They perfectly understood the numerical sequence from 1 to 9. However, they made mistakes in the other conditions (Cod1, Cod2, Cod3, and Over10) which contained the two-digit numerals. Processing the two-digit numerals was more difficult than the one-digit numerals. This result is congruent with the result of testing the factor of the range “1 to 9” vs. “1 to 19” (see Section 4.2).

The chimpanzee Pal is an interesting exception. She showed 100% accuracy in Over10. This result indicates that she is as good as humans at understanding the decimal number system and she scored 100% correct in the two-digit numerals (in the range 10 to 19). Pal’s performance was V-shaped, similar to humans. Chimpanzee Ai showed a similar tendency. In contrast, the accuracy of Cleo monotonically decreased as the number of two-digit numerals increased. The result corresponds to her poor performance in the assessment test of four adjacent numerals including the numeral 10 (see Section 4.3). Processing two-digit numerals are difficult for chimpanzees, but Pal shows the possibility of becoming close to human performance. However, a further detailed comparison of the two species is needed for the conclusion.

### 4.5. Comparison of Humans and Chimpanzees: Response Latency

Figure 12 shows individual data and averages for response latency (msec) in chimpanzees (*n* = 4) and humans (*n* = 5). The response latency was compared in the condition of four nonadjacent numerals in the range 1 to 19. Response latency is defined as the duration from the display onset to the first touch. Only correct trials were included in this analysis and the data from error trials were excluded.

All six human participants showed a monotonic increase in response latency as a function of the number of two-digit numerals, 0 through 4 (Figure 12, Right). The major difference between the five groups is the total number of digits to be processed (see Section 3.4). The more two-digit numerals, the more digits to be processed. The total number of digits that appeared on the screen increased from four, five, six, seven, and to eight at maximum along with the five conditions.

Participant H3 was the only person with 100% accuracy in all conditions. This was due to the unique style of taking a very long time before the execution, with no mistakes; thus, response latency was much longer than for the others. There was a tradeoff between accuracy and response latency. H3’s latency in each condition was extremely long at 5 to 6 s (see Section A.5: 4696 msec in Under10, 5425 msec in Cod1, 5891 msec in Cod2, 5735 msec in Cod 3, and 6505 msec in Over10). Despite longer latencies, the tendency was the same as the other five human participants. The latency simply increased as a function of the number of two-digit numerals, but as the values were outside the scale of the other participants, H3′s data were omitted from the analysis of response latency in Figure 12.

For chimpanzees, the data showed two patterns. Chimpanzees Ai and Pal followed the same tendency as humans: a monotonic increase in response latency as a function of the number of two-digit numerals 0 through 4. The more two-digit numerals, the more processing time is needed. Pal’s data were very similar to human data; her response latency increased from 492 to 1195 msec as a function of the two-digit numerals. Ai’s latency increased from 727 to 1328 msec. Their performance was similar to humans both in accuracy and response latency.

In contrast, the two chimpanzees Ayumu and Cleo deviated from the human tendencies. Ayumu’s latencies were flat regardless of the conditions, ranging from 594 to 672 msec. In the case of Cleo, latency was in the range of 554 to 859 msec and showed no monotonic increase. They did not take long to make a decision.

The comparison of response latency between naïve adult humans and chimpanzees shows a clear difference between the two species. All four chimpanzees were quicker than all six humans in all five conditions. Latency was about 700 msec for chimpanzees and about 1000 msec for humans on average. These results suggest that chimpanzees prioritize a quick response over a correct response. Their latencies remained low and more or less unchanged, even with deteriorating performance, whereas humans slowed down with increasing difficulty. This may indicate that chimpanzees make faster decisions than humans when processing the two-digit numerals in the visual discrimination task. However, we must acknowledge that the chimpanzees’ extensive, daily interactions with the device over several years may have had an impact on their motor skills, which in turn may have a positive effect on their response latencies.

### 4.6. Individual Differences in Chimpanzees

The six chimpanzee participants were not homogeneous. The basic notable information on individual differences follows. Ayumu (male) was almost at puberty. He enjoyed staying with estrus females and was sensitive to noise, such as screams and barks from the outside. Pan was not much attracted by the food reward, and sometimes did not eat it, especially in the hot summer season. She preferred to stay in the air-conditioned experimental room without participating in the trials. Interestingly, she could perform the task for social praise or with verbal encouragement.

Along with individual differences, two possible major influences on the results should be mentioned. One is the aging effect. The child chimpanzees always outperformed their mothers. As shown in Table 3, all three children were better than their mothers in baseline training. In the assessment test (see Table 4), average performances were: Ai 75% vs. Ayumu 85%, Chloe 61 % vs. Cleo 75 %, and Pan 57 % vs. Pal 79%, respectively. However, we did not run a statistical analysis because there were only three pairs.

The other influencing factor is shared “personality” within mother–child pairs. If we refer to families A (Ai and Ayumu), P (Pan and Pal), and C (Chloe and Cleo), family willingness to participate and perform in cognitive tests was in the order: A > P > C; again, however, we only mention this trend without statistical analysis.

## 5. Discussion

### 5.1. Sequence Order and Transitivity from Adjacent Numerals to Nonadjacent Numerals

The present study showed that chimpanzees can master the sequence order of numerals in the range 1 to 19 (see Section 4.1). Their understanding of the order was assessed by systematically manipulating four factors: range (1 to 9 vs. 1 to 19), adjacency (adjacent vs. nonadjacent numerals), number of numerals (3, 4, or 5 numerals), and memory (introducing the masking task). The study showed that training on adjacent numerals spontaneously transferred to nonadjacent numerals (see Section 4.2).

The baseline training provided the order of adjacent numerals. Chimpanzee Ai accurately touched each of the 16 simultaneously presented numerals from 1 to 16 in the VNM-Startfix task and the 11 numerals from 9 to 19 in the VNM-Endfix task. We never asked her to touch 19 numerals from 1 to 19 within a trial to avoid too much pressure on her. Interestingly, chimpanzee Ai invented new tactics to avoid some difficulties in the test. She understood the task nature of the VarNumMix tasks and made an “easy mistake” to finish the difficult trial quickly. She skipped it to move on to the next trial, which was expected to be easier. This “skipping behavior” may be an example of metaknowledge: knowing the nature of the task. 

The chimpanzees showed clear evidence of transitivity (see Section 4.2). The baseline training on touching adjacent numerals spontaneously transferred to nonadjacent numerals. This phenomenon is called “transitivity” or “transitive inference” [4,5,6,7,8,9], known to be within the capabilities of many animal species without training. Transitivity applies not only to cognitive tasks but also to social life [90]. For example, the social ranking order among nonhuman primates is called the dominance hierarchy [91,92,93,94]. Group-living animals do not have to learn about every possible social pair in their group but can infer the dominance order from witnessing a limited number of encounters. The evolutionary adaptation sensitive to this kind of order may provide the basis for a transitive inference of numerals in the present study.

### 5.2. Morphological Structure of the Decimal System in Terms of Visual Perception

The present study introduced the following 19 numerals: 1, 2, 3, 4, 5, 6, 7, 8, 9, 10, 11, 12, 13, 14, 15, 16, 17, 18, and 19 to the chimpanzees. The last nine two-digit numerals are morphologically similar from a visual perspective to the first nine one-digit numerals. If you disregard the left half of the complex numeral, the digit “1”, the sequential order of the numerals can be understood based on previous knowledge. The morphological feature of the decimal system in visual perception might help the chimpanzees in addition to the transitive inference in general.

The chimpanzees mastered the skill of the sequence of the Arabic numerals in the range 1 to 19, although the numeral 10 still caused difficulties for some chimpanzees such as Cleo, Chloe, and Pan (see Section 4.3). However, other chimpanzees mastered the skill very well, namely Ai, Ayumu, and Pal. They did not rely on rote memorization of the sequence, as they showed transfer to nonadjacent numerals (see Section 4.2). The transfer might be helped by the visual features of the decimal system. Further examination of the contribution of visual features (in other words, the morphology of the decimal system) was carried out by comparing humans and chimpanzees using the same procedure (see Section 4.4).

### 5.3. Difficulty in Processing the Numeral 10

The critical point at this stage was the numeral “10” and the introduction of the digit “0”. With the five numerals 5, 9, 12, 14, and 18, for example, you should not touch “12” first. You have to understand the two-digit numeral. Two-digit numerals should be ‘larger’ (positioned later in the sequential order) than one-digit numerals. Even more, the numeral “10” is located between 9 and 11. The numeral 0 has a crucial role in this regard.

The chimpanzee Ai had learned the meaning of “0” [53]. She mastered the ordinary scale of 0 to 9. She could understand and use “0” properly in naming tasks and choose “0” for nothing. She was fluent in both productive use and receptive use of numerals [54]. In short, Ai had acquired and established the number concept and utilized Arabic numerals to represent it. Additionally, because she had also learned the numeral 0, only Ai quickly learned the meaning of 10 [58]. She easily expanded her knowledge of the number “0 to 9” to “10 to 19” in the decimal system.

The other five chimpanzees never explicitly learned the meaning of “0”. However, the present study showed that their performance was approaching Ai’s thanks to the accumulated experience of the baseline training over the course of three years and eight months. At the second assessment test at the end of the present study, Ayumu outperformed Ai, and Pal was equal to Ai (see the summary in Table 3). This means that the chimpanzees did master the numerical sequence from 1 to 19 as Ai did, even without explicit training about the numeral 0. This appears reasonable when considering the acquisition of numerical sequences in human children. Without learning the meaning of 0 in advance, children can establish the numerical order including 10. They recite the sequence “one-two-three-four-five-six-seven-eight-nine-ten” without the concept of zero.

### 5.4. Global and Local: Dual Processing of Two-Digit Numerals

How do we understand the cognition of numerals in chimpanzees? The question can be framed as how chimpanzees perceive two-digit numerals in comparison to humans. The comparative experiment examined the performance when processing one-digit and two-digit numerals. Both accuracy and response latency showed characteristics shared by the two species. Processing two-digit numerals was harder than one-digit numerals. Here, we postulate that this result can be understood with reference to global and local processing (see Section 1) [62,63,64,65,66,67].

Global vs. local processing was originally devised by Gestalt psychologists. Navon (1979) published a paper titled: “Forest before trees: the precedence of global features in visual perception” [62]. Human perception is analytic and also holistic. The global precedence hypothesis (Navon effect) emphasizes the perceptual primacy of wholes. However, the comparative study of nonhuman primates, namely chimpanzees [68], baboons [69], and capuchin monkeys [70], shows perceptual primacy not in wholes but in local components. In other words, monkeys and apes may perceive each tree first and seldom think about the forest [68,95].

In our opinion, global–local processing might be an analog to the processing of visual words and letters in humans. Visual word recognition is a basic process involved in reading [96,97,98,99]. In the English language, a word contains some of the 26 letters of the Roman alphabet. There is a parallel processing of the global feature—the word—and the local feature—the letter [93]. We postulate that dual processing, such as word–letter, must exist in the visual processing of the two-digit numerals. Decimal numerals consist of 10 elementary digits of 0 to 9, and the combination makes an infinite number of integers, such as 13, 610, 2584, 10,946, and so on.

Because of the dual process of global-local, it takes more energy and time to recognize two-digit than one-digit numerals. The present study demonstrated that chimpanzees mastered the order of numerals in decimals in the range 1 to 19. However, the comparison with humans suggested difficulty in processing two-digit numerals in terms of global-local dual processing. In the cognitive tradeoff of accuracy and latency, humans take time for dual processing of both global and local features. Some chimpanzees such as Ai and Pal could do this too, while others such as Ayumu and Cleo could not. In general, all chimpanzees prefer to make quick decisions that focus on the local features of two-digit numerals. Humans’ proficiency in dual processing of global–local features results in them taking time to make the right decisions. There appears to be a cognitive tradeoff between the “local-quick” and “dual-slow” processing of global–local features [100]. The behavioral repertoire of touching the numerals from 1 to 19 may open a new window to understanding species-specific ways of information processing.

### 5.5. Individual Differences and Aging Effect

One major factor in the observed individual differences is age: child chimpanzees outperformed their mothers. This effect of aging was reported for other cognitive tasks in the same six chimpanzees, concerning auditory-visual position learning [101], sequential learning of one-digit numerals [2], and memory retention [1].

Another major factor was shared by the mother–child pairs. Mothers and their children showed similarities in both behavior and cognitive performances. For example, the C family (Chloe and Cleo) was characterized by quick decisions. Chimpanzees, like humans, possess broad intellectual capacities that are affected by their personalities [102,103,104,105,106]. These results highlight the importance of considering individual differences, including personality when evaluating responses in cognitive and behavioral tests. What kind of personality is involved may become a hot topic in future studies.

### 5.6. An Outlook for Future Studies

The present study used a sequential ordering task with Arabic numerals. One original aim was to understand the number concept in chimpanzees. What kind of numerosity judgments exist in animals: subitizing, counting, or magnitude estimation? The answer remains unclear, because the numerical repertoire of animals is small, with a few exceptions [42,43]. One solution was to extend the numerical sequence to produce a larger repertoire. This was accomplished in the present study by covering the range from 1 to 19. What follows should be the introduction of cardinal numbers to assess the numerosity judgment to many items up to 19. The introduction of zero (the digit “0”) in the numerical sequence must be an important step, as is extending two-digit numerals into the range of 20 to 29, 30 to 39, and so on. Is there any transfer from the experience of 1 to 19?

Working memory has been tested in the range 1 to 9, revealing the extraordinary memory of young chimpanzees [1,100]. The memory test can be extended to two-digit numerals. This was partly done in the assessment test which had five numerals; however, it became clear that processing two-digit numerals is not so easy for chimpanzees, and it is not purely a matter of working memory. Further direct comparisons of humans and chimpanzees may be needed in this field.

The results suggest that chimpanzees prioritize a quick response over a correct response. Their latencies remain short and more or less unchanged whereas humans slow down their response with increasing task difficulty. We concluded that there may be a cognitive tradeoff between “local-quick” vs. “dual-slow” processing of global–local features. If so, it would be interesting to study completely naïve humans, in which participants receive no verbal instruction for the task and only minimal instruction for the touchscreen device; thus, they would have to figure things out for themselves with auditory feedback/reward alone, similar to the chimpanzees. Such a study could track learning curves over time and see if humans still prioritize accuracy over solving the task quickly. For comparison, we have long-term chimpanzee data from the very early stages of their training. The comparison of completely naïve humans and chimpanzees could illuminate the evolutionary origins of human visual information processing.

To close the article, we mention the relevance of the study paradigm used here in terms of animal welfare. We used a twin-booth system (see Section A.6) designed to keep the mother-infant pairs free and comfortable: the child was not fully separated from their mother. However, as the children grow up, the twin booths can be utilized for studies of cooperation and communication [80,81,82,107,108,109,110]. The two booths can be separated and connected by electronic vertically sliding doors. The study of social intelligence in the twin booths together with the touchscreen system might be an important research method. The cognitive study of chimpanzees must be accompanied by the promotion of environmental enrichment. The key is the freedom to join the experiments [85,88].

## 6. Conclusions

The present study taught the numerical sequence from 1 to 19 to six chimpanzees: three pairs of mother and child. They had previous experience of touching the numerals 1 to 9 in ascending order. Then, the decimal number system was introduced and the chimpanzees learned to touch the numerals 1 to 19. The sequential touching was taught in two different conditions. One was to touch from 1 to the numeral X (Startfix condition). The other was to touch from the numeral X to 19 (Endfix condition). Daily baseline training was based on the two conditions of adjacent numerals. A systematic test examined the following four factors: range (1 to 9 vs. 1 to 19), adjacency (adjacent vs. nonadjacent), number of numerals (three, four, or five numerals), and memory (nonmemory task vs. memory task). All four factors were important. The narrow range (1 to 9) was easier than the wide range (1 to 19). Adjacent numerals were easier than nonadjacent numerals for expressing the ascending order. As the number of numerals increased, performance decreased. Memory tasks caused deterioration of performance. The further examination focused on processing the numeral 10. Performance was relatively low when the numeral 10 was involved. Taken together, with some difficulties, chimpanzees can master the sequence order in the range 1 to 19 in the two-digit Arabic numerals.

Direct comparison was carried out between humans and chimpanzees using the same apparatus and the same procedure. It was revealed that both humans and chimpanzees have relative difficulty processing two-digit numerals compared to one-digit numerals. The results were discussed in the framework of the global–local problem in information processing. Humans are good at processing multilevel information, including dual global and local levels. In contrast, chimpanzees tend to focus on local features and make quick decisions, much faster than humans. The difference might be due to the cognitive tradeoff of chimpanzee-like “local but quick” vs. human-like “dual but slow” information processing.

## Figures and Tables

**Figure 1 animals-13-00774-f001:**
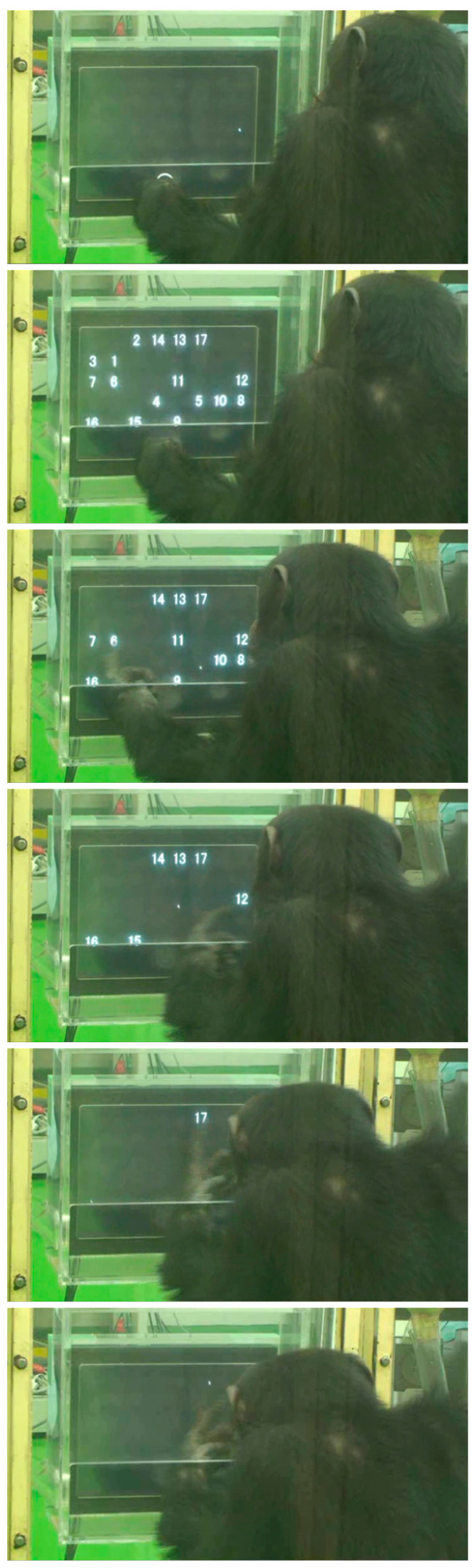
The touchscreen apparatus. It was encased in a translucent acrylic box and set just behind a translucent panel which prevented the chimpanzee from strongly banging the screen. The participant touched the screen through a window opened at the lower part of the box. Here, chimpanzee Ayumu is touching the numerals presented on the display in ascending order using his left index finger. The figure was cut out from the video clip (see Appendix A). The following video clip is available to the public: https://www.youtube.com/watch?v=cK-dtUTb8ME accessed on 6 December 2022.

**Figure 2 animals-13-00774-f002:**
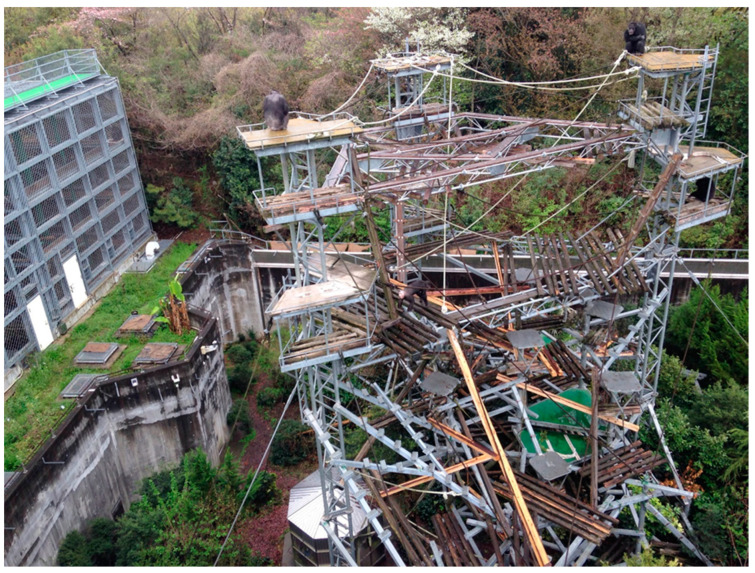
Chimpanzees’ outdoor enclosure at KUPRI. A group of 14 chimpanzees lived in an enriched environment. A participant chimpanzee came to the test booth based on their own free will.

**Figure 3 animals-13-00774-f003:**
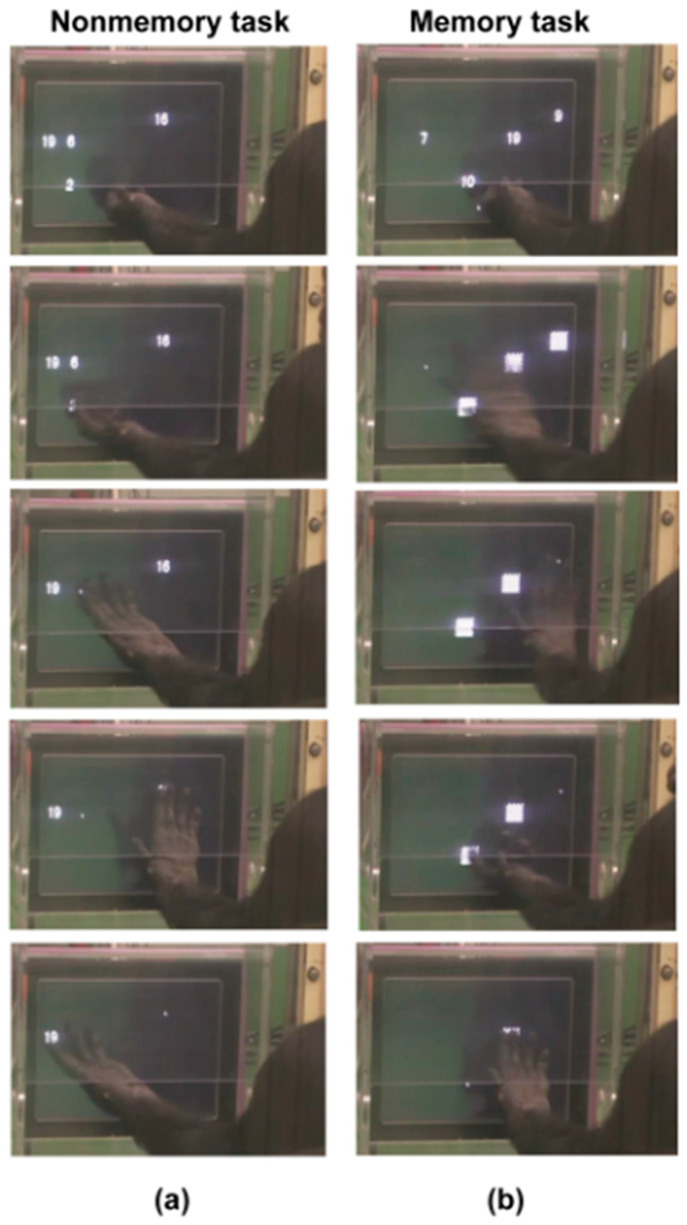
Assessment tests: range, adjacency, number of numerals, and memory. Chimpanzee Pal is touching the numerals in ascending order. (**a**) Four nonadjacent numerals in the range 1 to 19 in the nonmemory task. (**b**) Four nonadjacent numerals in the range 1 to 19 in the "masking" memory task. Individuals have different ways of touching numerals; Pal uses her left middle finger while keeping her palm up.

**Figure 4 animals-13-00774-f004:**
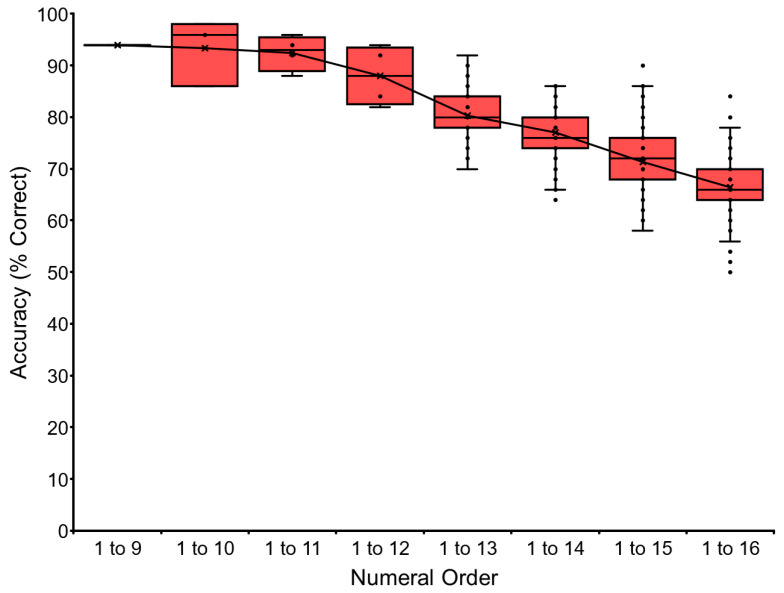
Data of chimpanzee Ai in the baseline training of VNM-Startfix task. The task started from VNM 1 to 9, in which the numerals appeared as either 1, 1-2, 1-2-3, 1-2-3-4, 1-2-3-4-5, 1-2-3-4-5-6, 1-2-3-4-5-6-7, 1-2-3-4-5-6-7-8, or 1-2-3-4-5-6-7-8-9. In the case of VNM 1 to 10, one more numeral sequence, 1-2-3-4-5-6-7-8-9-10, was added. The chimpanzee had to touch the numerals from 1 to X in ascending order. The X-axis showed the number of numerals presented at the maximum level in the task. The Y-axis showed the accuracy (% correct). The boxplot shows the quartile range of performance and the line shows the average accuracy.

**Figure 5 animals-13-00774-f005:**
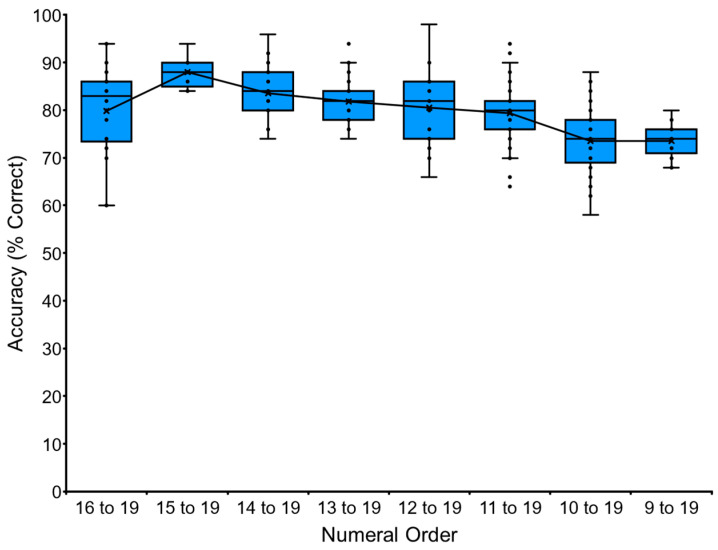
Data of chimpanzee Ai in the baseline training of the VNM-Endfix task. The chimpanzee had to touch the numerals from X to 19 in ascending order. Y-axis showed the accuracy (% correct). The boxplot shows the quartile range of performance and the line shows the average accuracy.

**Figure 6 animals-13-00774-f006:**
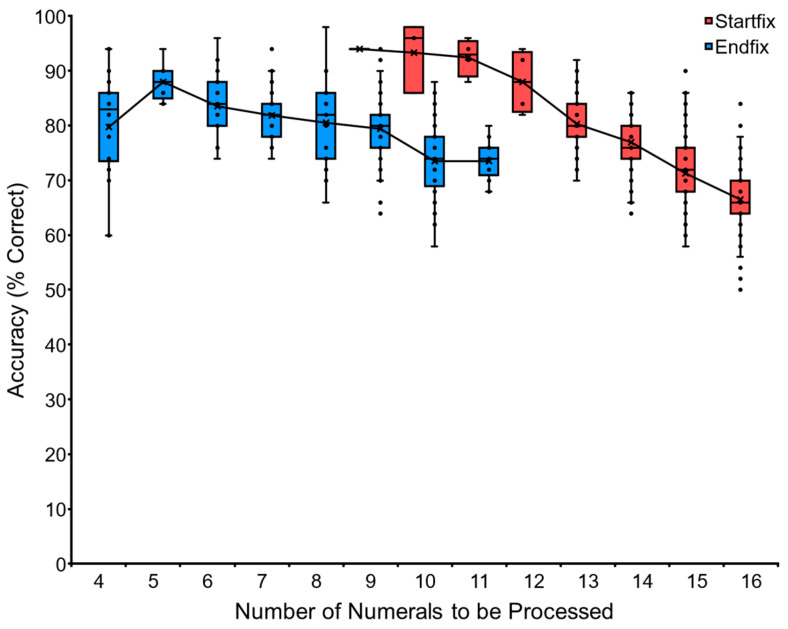
Data of chimpanzee Ai in the baseline training of two kinds of VNM tasks. In the VNM-Startfix task (red symbols), the numerals always started with 1. In the VNM-Endfix task (blue symbols), the numerals always ended in 19. In both cases, the chimpanzee had to touch the numerals in ascending order. Y-axis shows accuracy (% correct). Each dot shows performance in each session. Performance on the two tasks was plotted in terms of the maximum number of numerals (meaning the longest sequence) to be processed.

**Figure 7 animals-13-00774-f007:**
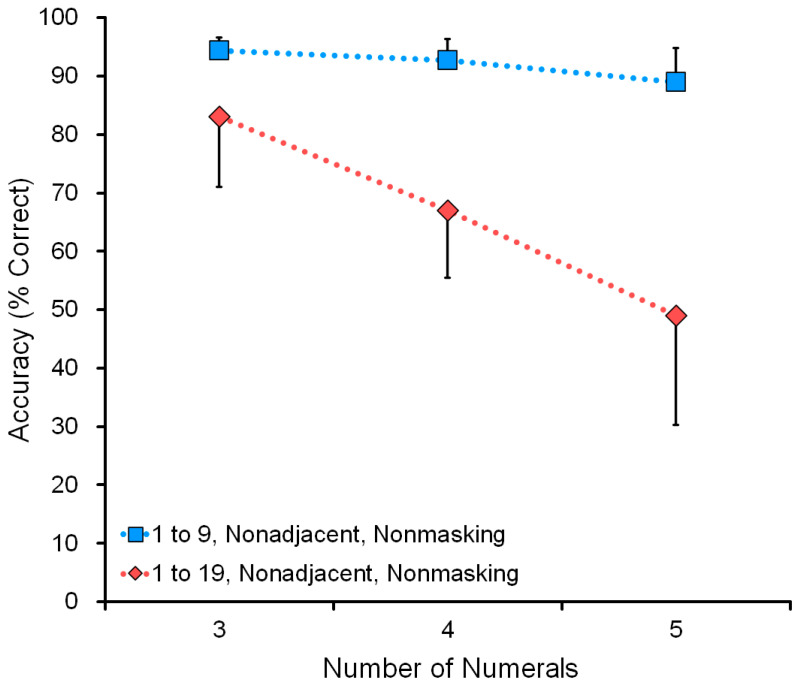
Data from all six chimpanzees were combined in the assessment test of range. The range was either “1 to 9” or “1 to 19”. Adjacency was kept as nonadjacent numerals. The X-axis (3, 4, and 5) shows the number of numerals presented on the screen. The Y-axis shows accuracy (% correct).

**Figure 8 animals-13-00774-f008:**
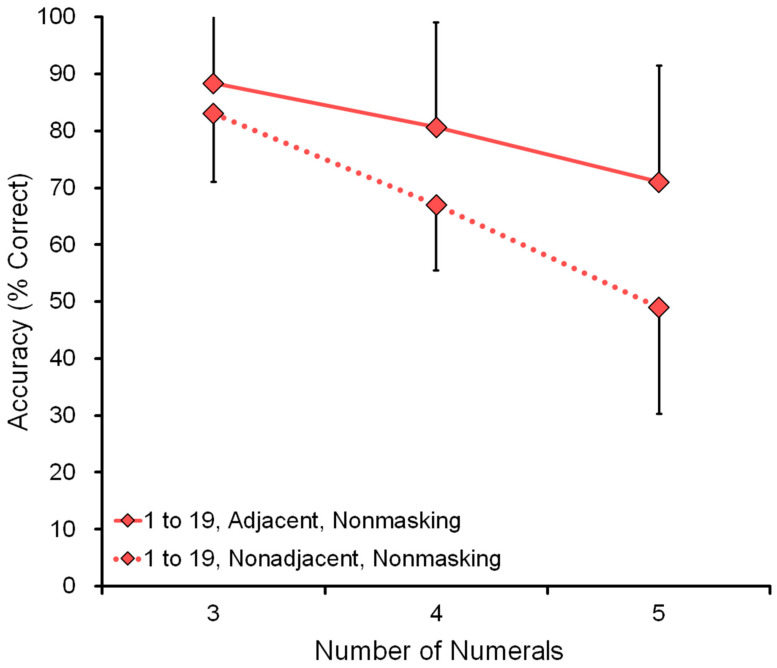
Data from all six chimpanzees were combined in the assessment test of adjacency. Either adjacent numerals (solid lines) or nonadjacent numerals (dotted lines) were presented. The range was from 1 to 19. The X-axis shows the number of numerals presented on the screen. The Y-axis shows accuracy (% correct).

**Figure 9 animals-13-00774-f009:**
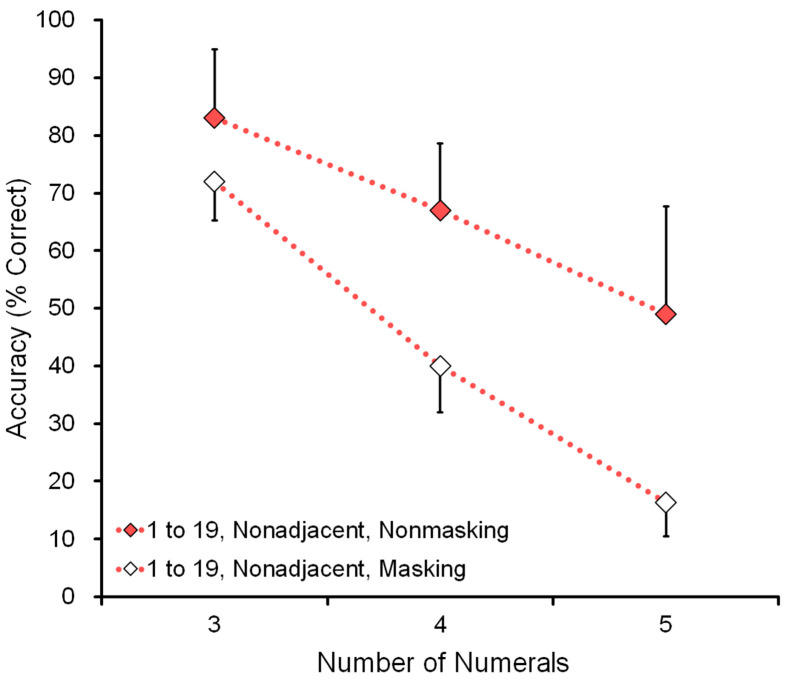
Data from all six chimpanzees were combined to show the factor of memory, i.e., whether the task was an ordinary nonmasking task (solid symbols) or the masking task (open symbols) that required memorizing the numerals. The X-axis shows the number of numerals, and the Y-axis shows accuracy (% correct). The condition is the range 1 to 19 and the nonadjacent numerals.

**Figure 10 animals-13-00774-f010:**
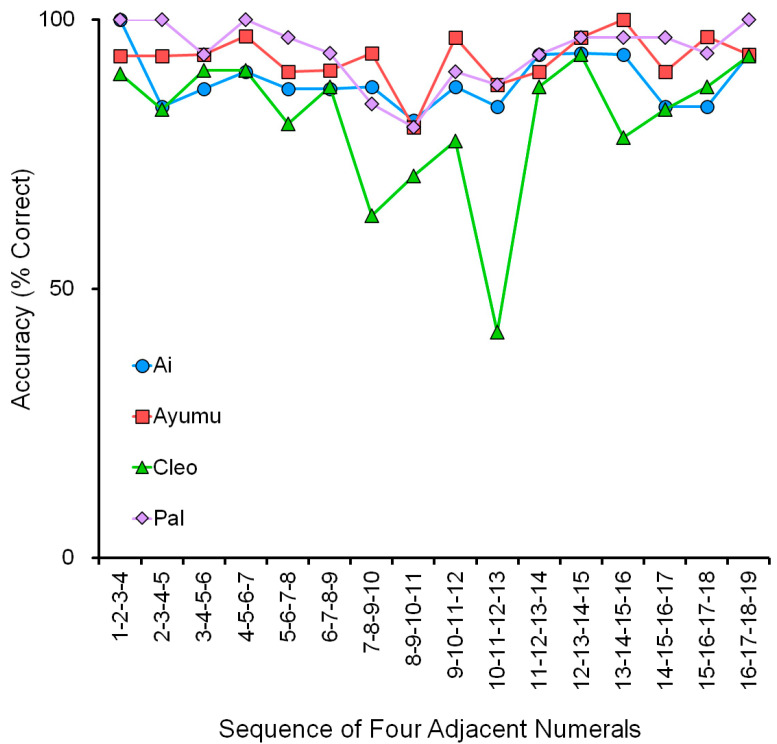
Individual data for four chimpanzees, showing the difficulty of ordering sequences containing the numeral 10. Each of the 16 patterns was tested 31 or 32 times in a total of 500 trials (see Section 3.3). Despite their intensive training in the sequence from 1 to 19, chimpanzees performed relatively poorly on patterns that included the numeral 10. Cleo’s data show that she found sequences containing the numeral 10 particularly difficult. The X-axis shows the sequence of four adjacent numerals, and the Y-axis represents accuracy (% correct).

**Figure 11 animals-13-00774-f011:**
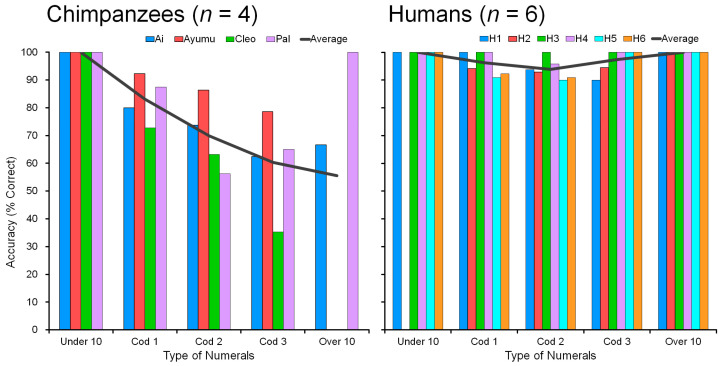
Comparison of chimpanzees (Left) and humans (Right) in the task of touching four nonadjacent numerals in the range 1 to 19. Individual accuracy data (% correct) are plotted in the bar graph, with average performance in the solid lines. The X-axis shows the conditions of the four presented numerals (see main text). Human performance shows a V-shape: performance on all one-digit and all-two-digit numerals was perfect, but performance on the in-between conditions deteriorated. Chimpanzee performance deteriorated as a function of the number of two-digit numerals, but Pal’s performance was V-shaped, similar to humans. Pal showed 100% in the Over10 condition, just like humans. In contrast, Cleo’s performance monotonically decreased and showed 0% accuracy in the Over10 condition. There are two missing data: data for Ayumu in the Over10 condition and H2 in the Under10 condition are unfortunately missing because of the purely randomized procedure. These two extreme conditions of Under10 and Over10 had very low occurrence in a test session of 50 trials. A total of 3876 possible patterns of 4 numerals were purely randomized in this test. Thus, there happened to be no trials that matched these conditions in the particular participants. However, the general trend is preserved.

**Figure 12 animals-13-00774-f012:**
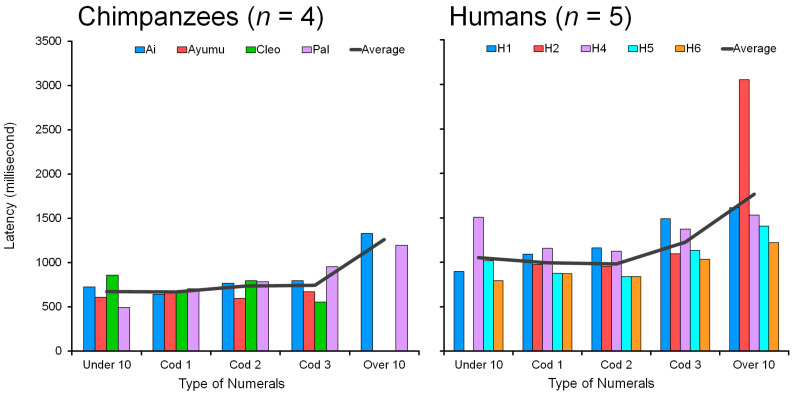
Comparison of chimpanzees (Left, *n* = 4) and humans (Right, *n* = 5) in the task of touching four nonadjacent numerals in the range 1 to 19. Individual response latencies (in msec, Y-axis) are plotted in the bar graph, and average performance is shown in the solid lines. The X-axis shows the conditions of the four presented numerals, as described in Section 3.4 and Figure 11. Chimpanzees responded faster than humans. In humans, the latency for all one-digit numerals was about 1000 msec, but for all two-digit numerals, it increased to about 1500 msec. In chimpanzees Ayumu and Cleo, the latency remained constant at about 700 msec throughout the conditions, much shorter than in naïve adult humans. Ai and Pal showed a similar tendency to humans. There were three missing data because of the purely randomized procedure: there happened to be no trials or no correct trials that matched these conditions in the particular participants (Ayumu, Cleo, and H2). Participant H3′s value was well outside the range of the other participants (see Section A.5), so H3′s data were omitted from this graph and further analysis of response latency. The exact latencies of all six humans including H3 are shown in the Appendix A.

**Table 1 animals-13-00774-t001:** List of participant chimpanzees in KUPRI. Age is at the beginning of the present study. GAIN stands for Great Ape Information Network, which is a database of all chimpanzees living in Japan (see: https://shigen.nig.ac.jp/gain/LocaleAction.do accessed on 6 December 2022). GAIN is equivalent to ChimpCare in the USA (see: https://chimpcare.org/map/ accessed on 6 December 2022). “Numerals experience” means the experience before starting the learning of two-digit numerals.

Name	Sex	GAIN ID	Mother/ID	Birth	Age	Numerals Experience
Ai	female	0434	na	1976?	34	0–9
Chloe	female	0441	Charlotte/na	12 December 1980	30	1–9
Pan	female	0440	Puchi/0436	7 December 1983	27	1–9
Ayumu	male	0608	Ai/0434	24 April 2000	10	1–9
Cleo	female	0609	Chloe/0441	19 June 2000	10	1–9
Pal	female	0611	Pan/0440	9 August 2000	10	1–9

**Table 2 animals-13-00774-t002:** Summary of training tasks and assessment tests. All six chimpanzees received training and tests in the same chronological order. However, the progress was different among chimpanzees due to their availability, their willingness to participate, and their learning speed. The first row shows the number of days on which chimpanzees came to the test booths. Other cells represent the number of sessions required for each stage. “na” means not applicable, i.e., not done for that particular chimpanzee. All sessions consisted of 50 trials.

									Chimpanzees					
Section	Task Name, Abbreviation, and Definition								Ai	Ayumu	Chloe	Cleo	Pan	Pal
	*Total days of coming to the test booth in the study period*								630	630	477	477	498	498
3.1.	** *Baseline training of touching adjacent numerals:* ** ** *VarNumMix (VNM) task in the range of 1 to 19* **													
		VNM task which characterized by the “Startfix” condition												
		The sequence always started from the numeral 1												
				VNM-Startfix 1 to 9					1	1	14	46	16	3
				VNM-Startfix 1 to 10					3	3	19	23	18	8
				VNM-Startfix 1 to 11					4	4	37	35	165	3
				VNM-Startfix 1 to 12					4	13	217	43	254	20
				VNM-Startfix 1 to 13					97	17	221	41	na	55
				VNM-Startfix 1 to 14					117	108	na	178	na	77
				VNM-Startfix 1 to 15					300	46	na	116	na	108
				VNM-Startfix 1 to 16					151	109	na	na	na	18
				VNM-Startfix 1 to 17					na	151	na	na	na	190
				VNM-Startfix 1 to 18					na	136	na	na	na	na
				VNM-Startfix 1 to 19					na	na	na	na	na	na
				total (number of sessions)					677	588	508	482	453	482
		VNM task which characterized by the “Endfix” condition												
		The end of the sequence was always fixed as 19												
				VNM-Endfix 16 to 19					22	3	56	12	56	39
				VNM-Endfix 15 to 19					9	4	25	7	157	3
				VNM-Endfix 14 to 19					15	3	90	9	39	4
				VNM-Endfix 13 to 19					26	9	72	24	na	12
				VNM-Endfix 12 to 19					22	9	na	80	na	16
				VNM-Endfix 11 to 19					52	16	na	98	na	74
				VNM-Endfix 10 to 19					92	28	na	8	na	50
				VNM-Endfix 9 to 19					17	125	na	na	na	148
				VNM-Endfix 8 to 19					na	8	na	na	na	na
				VNM-Endfix 7 to 19					na	na	na	na	na	na
				total (number of sessions)					255	205	243	238	252	346
3.2.	** *First Assessment tests (4 factors):* ** ** *range, adjacency, number of numerals, and memory* **								24	24	24	24	24	24
3.2.	** *Second Assessment tests (4 factors):* ** ** *range, adjacency, number of numerals, and memory* **								24	24	24	24	24	24
3.3.	** *Assessment of adjacent numerals including the numeral 10* **								10	10	na	10	na	10
		The task was 4 *adjacent* numerals in the range of 1 to 19 with nonmemory task												
3.4	** *Comparison of humans and chimpanzees* **								12	12	12	12	12	12
		The task was 3,4, or 5 *nonadjacent* numerals in the range of either 1 to 9 or 1 to 19 with nonmemory task												

Notes: Assessment tests (4 factors): The 1st test was done in September 2013 and 2nd test in March 2014; na: not applicable.

**Table 3 animals-13-00774-t003:** Summary of accuracy (% correct) in baseline training tasks (VNM tasks) in each chimpanzee. Each cell represents a condition and a chimpanzee. The tables show average performances. “na” means not applicable, i.e., not done for that particular chimpanzee.

Task Name						Chimpanzees						
		Condition				Ai	Ayumu	Chloe	Cleo	Pan	Pal	Average
VNM task which characterized by the “Startfix” condition												
The sequence always started from the numeral 1												
		VNM-Startfix 1 to 9				94	92	79	80	82	83	85
		VNM-Startfix 1 to 10				93	96	82	81	84	88	87
		VNM-Startfix 1 to 11				93	92	80	81	80	88	86
		VNM-Startfix 1 to 12				88	84	71	76	68	83	78
		VNM-Startfix 1 to 13				80	77	65	71	na	78	74
		VNM-Startfix 1 to 14				77	80	na	67	na	78	76
		VNM-Startfix 1 to 15				71	80	na	61	na	74	72
		VNM-Startfix 1 to 16				66	79	na	na	na	69	71
		VNM-Startfix 1 to 17				na	77	na	na	na	63	70
		VNM-Startfix 1 to 18				na	63	na	na	na	na	63
		VNM-Startfix 1 to 19				na	na	na	na	na	na	
		Average accuracy in total				83	82	75	74	79	78	78
VNM task which characterized by the “Endfix” condition												
The end of the sequence was always fixed as 19												
		VNM-Endfix 16 to 19				80	97	65	72	73	69	76
		VNM-Endfix 15 to 19				88	97	80	88	72	94	87
		VNM-Endfix 14 to 19				84	95	80	88	61	91	83
		VNM-Endfix 13 to 19				82	86	72	79	na	84	81
		VNM-Endfix 12 to 19				81	88	na	75	na	84	82
		VNM-Endfix 11 to 19				79	84	na	73	na	79	79
		VNM-Endfix 10 to 19				74	80	na	63	na	75	73
		VNM-Endfix 9 to 19				74	78	na	na	na	76	76
		VNM-Endfix 8 to 19				na	65	na	na	na	na	65
		VNM-Endfix 7 to 19				na	na	na	na	na	na	
		Average accuracy in total				80	86	74	77	69	82	78

**Table 4 animals-13-00774-t004:** Data for all six chimpanzees in the second assessment test. Four factors influenced numerical ordering performance. First, the range of numerals was either 1 to 9 or 1 to 19. Second, the adjacency was either adjacent or nonadjacent. Third, ‘memory’ means whether the task was an ordinary task (nonmasking) or the masking task, which required the memorizing of numerals. Fourth, the number of numerals was either three, four, or five. In all conditions, the chimpanzee had to touch the numerals from 1 to 9 or from 1 to 19 in ascending order. Each cell shows accuracy (% correct). These data are from the second test; corresponding data from the first test performed six months earlier are available in Section A.4.

Task				Chimpanzee Participants						
Range	Adjacency	Memory	Number of Numerals	Ai	Ayumu	Chloe	Cleo	Pan	Pal	Average
1–9	Adjacent	Nonmask	3	98	98	94	98	90	98	96.0
			4	96	90	86	92	88	98	91.7
			5	88	90	78	90	82	94	87.0
		Mask	3	90	100	76	92	78	98	89.0
			4	74	88	60	88	42	90	73.7
			5	62	90	22	68	24	62	54.7
	Non-adj	Nonmask	3	94	96	92	98	92	94	94.3
			4	96	96	92	92	86	94	92.7
			5	92	94	92	78	88	90	89.0
		Mask	3	94	100	82	90	80	84	88.3
			4	86	100	60	82	42	80	75.0
			5	54	86	36	58	8	66	51.3
1–19	Adjacent	Nonmask	3	96	98	76	98	64	98	88.3
			4	86	100	54	82	64	98	80.7
			5	76	92	50	78	42	88	71.0
		Mask	3	88	98	46	94	80	92	83.0
			4	68	86	30	74	42	78	63.0
			5	34	50	24	44	8	48	34.7
	Non-adj	Nonmask	3	90	96	64	78	78	92	83.0
			4	72	86	62	54	58	70	67.0
			5	52	84	36	36	36	50	49.0
		Mask	3	72	76	76	66	62	80	72.0
			4	34	48	48	44	28	38	40.0
			5	14	8	18	16	16	26	16.3
			Average	75	85	61	75	57	79	**72.1**

## Data Availability

All of the fundamental data sets that we analyzed are available on the following site in Figshare Dataset: Muramatsu, Akiho; Matsuzawa, Tetsuro (2023): Arabic numerals 1 to 19 in chimpanzees. figshare. Dataset. https://doi.org/10.6084/m9.figshare.22121345.v1. The video data is available: https://doi.org/10.6084/m9.figshare.22124210. Further detailed data are available from the corresponding author upon reasonable request.

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
