# Peer review of "Sequence Order in the Range 1 to 19 by Chimpanzees on a Touchscreen Task: Processing Two-Digit Arabic Numerals"

_animals, 2023, doi:10.3390/ani13050774_

Round 1

Reviewer 1 Report

The present study compares numerical sequence learning of 2 digit-numbers in 3 chimpanzee mother-child dyads and in 6 humans. The authors propose to interpret the difference between  the two species in the framework of global versus local information processing. The study is part of the Japanese program GAIN about ape cognition. These are rare and precious data that must be widely shared with the rest of the primate community. They fit perfectly with the topic of the special issue Recent Advances in Animal Cognition and Ethology and Animals is the right journal for these valuable data.

The paper is well written, the methods are sound and the results clearly presented. I have but only minor comments and questions.

1. Chimpanzee mother-child dyads were tested together in adjacent booths. I may have missed the information but were humans also tested in pairs? Were chimpanzees tested in pairs only to speed up data collection or because mother and child cooperated more willingly to testing in duo than in solo?

2. Can the authors provide some speculation about possible links between the apes' personality traits and their individual  talent in sequence learning?

3. What about non-numerical sequence learning such as the learning of sequences of actions for example? How do the present results fit with this other literature about sequential learning? Would the global-local information processing hypothesis apply as well?

Reviewer 2 Report

Please see report for general and detailed comments. Thank you.

Round 2

Reviewer 2 Report

Thank you very much for addressing all the points I made in my first review in such a thorough and extensive manner. I'm generally happy with the changes made. However, one aspect is still of great concern to me and I would like to have this addressed (I added a file with more explanation). Further, I recommend doing another check for grammar and spelling of the newly added texts. I think some improvement can be made to improve clarity/understanding and readability.

Author Response

Please see the attached file for the point-by-point response to the reviewer's comments.  Thank you.
